# Assessing geochemical and natural radioactivity impacts of Hamadat phosphatic mine through radiological indices

Douaa Fathy[1], Hesham M. H. Zakaly[2,3,4]*, El Saeed R. Lasheen[5], Reda Elsaman[2], Saad S. Alarifi[6], Mabrouk Sami[1], Hamdy A. Awad[7], Antoaneta Ene[8]*

1 Faculty of Science, Geology Department, Minia University, El-Minia, Egypt, 2 Faculty of Science, Physics Department, Al-Azhar University, Assiut Branch, Cairo, Egypt, 3 Faculty of Engineering and Natural Sciences, Computer Engineering Department, Istinye University, Istanbul, Turkey, 4 Institute of Physics and Technology, Ural Federal University, Yekaterinburg, Russia, 5 Faculty of Science, Geology Department, Al-Azhar University, Cairo, Egypt, 6 Department of Geology and Geophysics, College of Science, King Saud University, Riyadh, Saudi Arabia, 7 Faculty of Science, Geology Department, Al-Azhar University, Assiut Branch, Cairo, Egypt, 8 Faculty of Sciences and Environment, Department of Chemistry, Dunarea de Jos University of Galati, INPOLDE Research Center, Physics and Environment, Galati, Romania

* h.m.zakaly@gmail.com, h.m.zakaly@azhar.edu.eg (HMHZ); Antoaneta.Ene@ugal.ro (AE)

**Data Availability Statement:** All data generated or analyzed during this study are included in this article.

## Abstract

The utilization of phosphorite deposits as an industrial resource is of paramount importance, and its sustainability largely depends on ensuring safe and responsible practices. This study aims to evaluate the suitability of phosphorite deposits for industrial applications such as the production of phosphoric acid and phosphatic fertilizers. To achieve this goal, the study meticulously examines the geochemical characteristics of the deposits, investigates the distribution of natural Radioactivity within them, and assesses the potential radiological risk associated with their use. The phosphorites are massive and collected from different beds within the Duwi Formation at the Hamadat mining area. They are grain-supported and composed of phosphatic pellets, bioclasts (bones), non-phosphatic minerals, and cement. Geochemically, phosphorites contain high concentrations of $P_2O_5$ (23.59–28.36 wt.%) and CaO (40.85–44.35 wt.%), with low amounts of $Al_2O_3$ (0.23–0.51 wt.%), $TiO_2$ (0.01–0.03 wt.%), $Fe_2O_3$ (1.14–2.28 wt.%), $Na_2O$ (0.37–1.19 wt.%), $K_2O$ (0.03–0.12 wt.%), and MnO (0.08–0.18 wt.%), suggesting the low contribution of the detrital material during their deposition. Moreover, they belong to contain enhanced U concentration (55–128 ppm). They are also enriched with Sr, Ba, Cr, V, and Zn and depleted in Th, Zr, and Rb, which strongly supports the low detrital input during the formation of the Hamadat phosphorites. The high Radioactivity of the studied phosphorites is probably due to the widespread occurrence of phosphatic components (e.g., apatite) that accommodate U in high concentrations. Gamma spectrometry based on NaI (Tl) crystal 3×3 has been used to measure occurring radionuclides in the phosphorite samples. The results indicate that the radioactive concentrations' average values of $^{226}$Ra, $^{232}$Th, and $^{40}$K are 184.18±9.19, 125.82±6.29, and 63.82±3.19 Bq Kg$^{-1}$, respectively. Additionally, evaluations have been made of the radiological hazards. The calculated risk indicators exceeded the recommended national and world averages.

**Funding:** This research was supported by Researchers Supporting Project number (RSP2023R496), King Saud University, Riyadh, Saudi Arabia. The author AE would like to thank the support of the research grant with contract no. 9187/2023, funded by Dunarea de Jos University of Galati, Romania. The work of author HZ is covered from the Ministry of Science and Higher Education of the Russian Federation (Ural Federal University Program of Development within the Priority-2030 Program).

**Competing interests:** NO authors have competing interests.

The data obtained will serve as a reference for follow-up studies to evaluate the effectiveness of the Radioactivity of phosphatic materials collected from the Hamdat mine area.

## 1. Introduction

Radionuclides naturally spread throughout the earth because these nuclides are present in various geological materials such as rocks, soil, sand, water, coal, and phosphate deposits [1–3]. The Radioactivity in the scale samples comes from the $^{238}$U to $^{232}$Th series (and their decay progeny) as well as $^{40}$K. These radioisotopes play a large role in a person's dose through external and internal exposure due to ingestion or inhalation [4–8]. Phosphates have many stable and radioactive elements that can be of environmental interest to the public [4]. The risks arise because of the release of dust and polluting residues into the environment from phosphate rock when used in industrial facilities, primarily in producing phosphoric acid and fertilizers [4]. It's worth looking at the elemental concentration in phosphate ores because of the rising use of phosphate in industries worldwide. It is used as a P-fertiliser and phosphorous source for chemical and food industries [5].

In Egypt, there is a wide occurrence of igneous, metamorphic, and sedimentary rocks which are concentric essentially in Sinai, the Eastern, and Western Deserts [7,9,10]. Phosphate deposits in Egypt are part of the Duwi Formation, which stretches from the Arab countries in Africa (to the west) to the Arab countries in Asia (to the east). According to [11], Egyptian phosphorite occurrences can be separated into three belts that tend in an east-west direction. The economic occurrences are only found in the central facies band and are confined to the following localities: the coast of the Red Sea from Safaga to Quseir; the Nile Valley between Idfu and Qena; and the Western Desert between the Kharga and Dakhla oases (Abu Tartur area). Age estimates [11,12] place the Phosphorite (Duwi) Formation somewhere in the range of the Upper Campanian to the Early Maastrichtian. This formation is conformably overlain by the Dakhla Formation.

Marine sedimentary phosphorites, the world's primary supply of phosphorus fertilizers, are formed intermittently along continental and oceanic basin margins. Egypt's phosphate resources are part of the Upper Cretaceous-Lower Paleogene Tethyan phosphonic belts that run through North Africa and the Middle East [13,14]. The economic ore beds are distributed along the Red Sea Coast, the Western Desert, and the Nile Valley. They originated in basins of intracratonic sediment in an epicontinental sea on the stable African shelf [15]. They were linked to the most promising black shale source rocks formed in a warm greenhouse climate [16]. No study has been carried out on the Hamdat area within the Eastern Desert to study their geochemistry, natural Radioactivity, and potential radioactive contamination of phosphate rocks. Thus, the current work aims to measure the concentrations of ($^{232}$Th, $^{226}$Ra, and $^{40}$K) and radiological hazards for populations in samples of phosphates collected from the Hamadat mine area (Central Eastern Desert, Egypt). It is certain that the data extracted from this study is helpful for mapping natural Radioactivity and can also be used as reference data for monitoring and identifying potential radioactive contamination in the future. Additionally, the findings of this study will contribute to the development of best practices for industrial applications involving phosphate rocks in neighboring countries that have similar phosphate deposits.

In this manuscript, we provide geochemical data, measured natural Radioactivity, and radiological risk parameters of the Hamadat mine phosphorites to study their geochemical characteristics and evaluate their radiological risk.

## 2. Materials and methods

### 2.1. Whole-rock geochemical analysis

Phosphates were obtained in eleven samples from the Hamadat mining region (Central Eastern Desert, Egypt). The selected phosphorite samples were prepared for geochemical analysis. The samples were powdered using a mortar grinder in the Geology Department, Faculty of Science, Assiut University (Egypt). The major oxides (fused pellets) and trace elements (powder pellets) of the whole rock were measured using the Philips PW 24004, an X-Ray Fluorescence (XRF) technique at the University of Vienna (Austria). The details of the analytical steps were described by Ali et al. [17]. The XRF analytical precision was estimated to be better than 0.5% for major oxides and 2% to 5% for trace elements.

### 2.2. Measurements and analysis using gamma spectrometry

The rest of the samples were crushed, homogenized, and sieved (200 m), which is the ideal size and enriched in heavy minerals after being dried at 110˚C for 24 hours to remove the moisture content altogether. A 250 mL polyethene beaker served as the container for each sample. For four weeks, the beakers were sealed to establish secular equilibrium, which occurs when the daughters' decay rate reaches parity with that of the parents. This step's significance is to ensure that the daughters will remain in the sample and that the radon gas is contained within the volume [18,19].

The samples were examined using a high-resolution scintillation detector made of a 3-inch NaI (Tl) crystal in a gamma ray spectrometer. A 100 mm thick cylindrical lead shield with a fixed bottom and a moveable cover covered the detector to lower the gamma-ray background. Using reference sources such as $^{60}$Co (1173.2 and 1332.5 keV), 133Ba (356.1 keV), and 137Cs (661.9 keV), the detection array's energy was calibrated. The IAEA-314 reference material, whose particular activity was known and contained the radionuclides $^{226}$Ra, $^{232}$Th, and $^{40}$K, was used to create the efficiency calibration curve. An empty sealed beaker with the same geometry as the measured samples was placed around the detector to measure the background activity in order to adjust the net peak area of the detected isotopes. Twenty-eight thousand eight hundred seconds were used to measure the activity and the background. The average concentrations of $^{212}$Pb (238.6 keV) and $^{228}$Ac (911.1 keV) in the samples were used to calculate the concentration of $^{232}$Th, and the average concentrations of the decay products of $^{214}$Pb (351.9 keV) and $^{214}$Bi (609.3 and 1764.5 keV) were used to calculate the concentration of $^{226}$Ra. The 1461 keV peak was used to estimate the $^{40}$K activity [20,21]. Referring to the IAEA reports, the lower limit of detection (LLD) is given by

$$\text{LLD} = \frac{4.66\sqrt{F_c}}{\eta.P_\gamma.m.t} \tag{1}$$

Where Fc is the Compton background in the region of the selected gamma line in the sample spectrum, ε is the system detection efficiency, Pγ is the absolute transition probability of gamma decay, m is the sample mass (in kilograms), and t is the counting time (in seconds). The lower limits of detection (LLD) for $^{226}$Ra, $^{232}$Th, and $^{40}$K in samples are 2.4, 1.4, and 5.8 Bq kg$^{-1}$, respectively.

The activity concentration (A), the activity of radium equivalent (Ra$_{eq}$), absorbed dose rate (D), external and internal hazards indices (Hex, Hin), annual effective dose equivalent (AED), and excess lifetime cancer (ELCR), was calculated by the following equations [22–25]:

$$A = \frac{N_p}{e \times \eta \times m} \tag{2}$$

Where NP is the count per second, e is the abundance of the gamma peak in a radionuclide, η is the measured efficiency for each gamma peak observed for the same number of channels, either the sample or calibration source and m is the sample mass.

$$\text{Raeq} = \text{ARa} + (1.43\text{ATh}) + (0.077\text{AK}) \tag{3}$$

$$\text{D} = 0.427\,\text{ARa} + 0.662\text{ATh} + 0.0423\,\text{AK} \tag{4}$$

$$\text{Hex} = \frac{A_{Ra}}{370} + \frac{A_{Th}}{259} + \frac{A_K}{4810} \tag{5}$$

$$\text{Hin} = \frac{A_{Ra}}{185} + \frac{A_{Th}}{259} + \frac{A_K}{4810} \tag{6}$$

Where, ARa, ATh, and AK are the activities of $^{226}$Ra, $^{232}$Th, and $^{40}$K, respectively, in Bq kg 1.

$$\text{Annual Effective Dose Rate} = \text{D} \times \text{T} \times \text{F} \tag{7}$$

Where, T is the outdoor occupation time (0.224 h365.25 d1753 hy$^{-1}$), D is the dose rate (in nGyh$^{-1}$), F is the conversion factor (0.7 10–6 SvGy-1),

$$\text{ELCR} = (\text{AED}) \times \text{LD} \times \text{RF} \tag{8}$$

Where DL is the life expectancy (70 years), and RF is the risk factor (Sv-1), which is 0.05. (ICRP-60).

## 3. Geologic background and petrography

In the Eastern Desert, the phosphate beds occur in the Duwi Formation. The Duwi Formation is overlain by laminated foraminiferal-rich marine shale and marl of the Late Formation of Maastrichtian Dakhla, which is conformably underlain by non-marine, the middle Campanian Quseir Formation-varicolored shale [26]. The Duwi deposits depict the first stage of marine transgression during the Late Cretaceous. These sediments indicate deposition in settings ranging from the inner neritic to the outer shelf, as well as recurrent sea-level variations [27]. Vertebrate remains, fish teeth, coprolites, heteromorph ammonites, and bivalves, as well as a few gastropods, echinoderms, baculitid ammonites, and nautiloid species, are recorded in this formation [28,29]. Based on the faunal assemblage in this formation, these deposits are mostly restricted to the Late Campanian-Early Maastrichtian time [14,30]. The Duwi Formation is occupied by alternating strata of sandstone, claystone, siltstone, cherts, black shales, and oyster limestone that enclose a series of varying thicknesses of phosphatic interbed shales [26].

The Hamadat phosphatic mine area is bounded by longitudes 34° 11′ 25′′-34°11′ 32′′E and latitudes 26°02′ 55′′-26° 03′ 2′′N in the Central Eastern Desert (Fig 1A). Fig 1 was generated utilizing CorelDRAW software for the creation of the map image. The area can be easily reached through the Qift-Quseir asphaltic road about 16 km west. At the Hamadat mine area, the phosphates were primarily collected from the Duwi Formation horizontal strata (Fig 1B), which extended to several kilometres and were deposited during late Cretaceous marine transgression events [30]. The phosphates occur in the form of beds that interact with shale, limestone, and chert (Fig 2A–2F).

The phosphorite samples are massive, fine- to medium-grained with grey, and yellow-to-brown colours (Fig 3A and 3B). The studied phosphorites belong to the grain-supported type

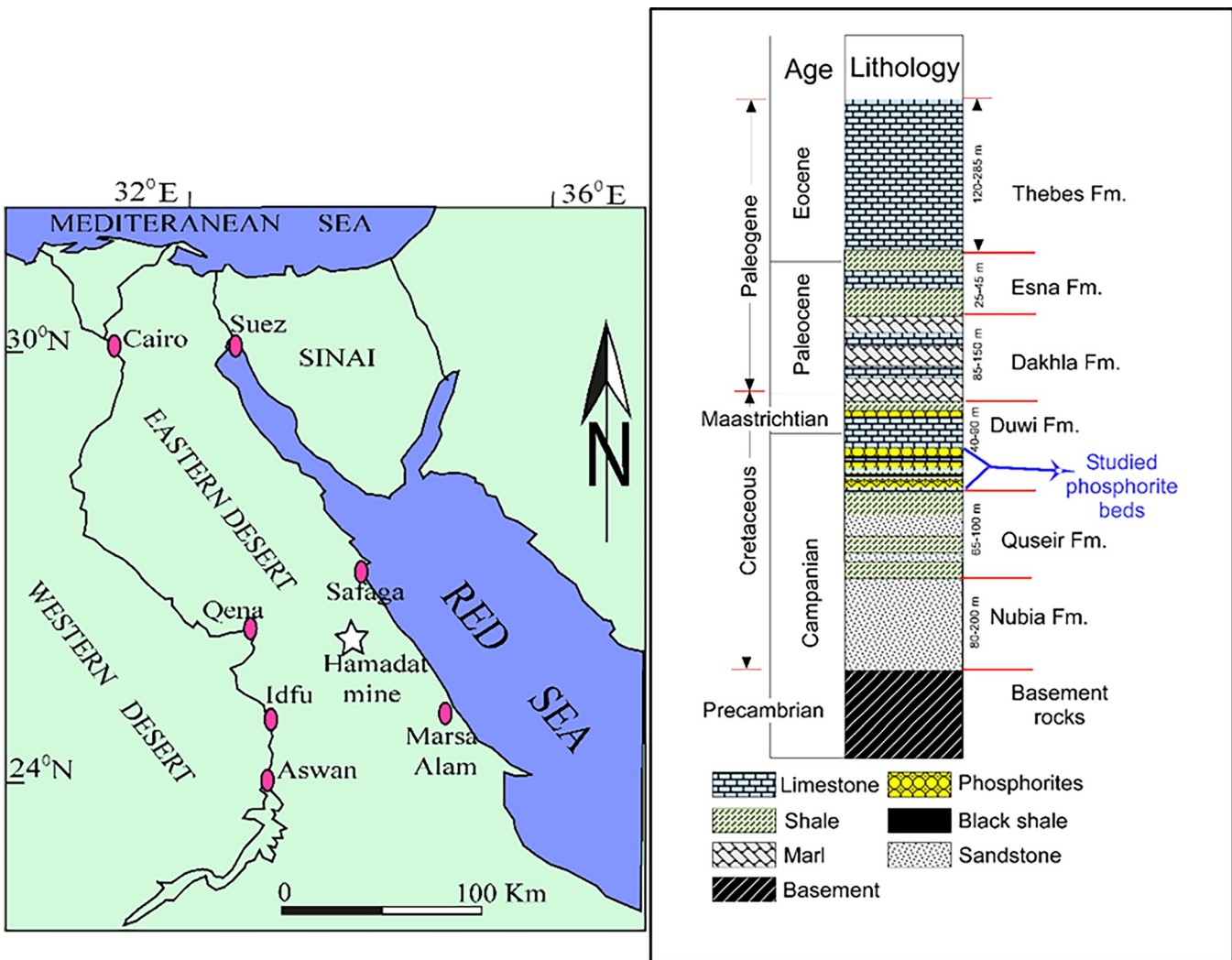

**Fig 1.** a) Location map for the investigated Hamadat area in the Eastern Desert, and (b) Lithostratigraphic column for the studied area.

where the phosphatic components exceed 70 vol% of the rock. The phosphorite samples collected from the upper layers (1.5 m thickness) are grey and fossiliferous. At the same time, those collected from the medium and lower layers have common yellowish to brownish colours. They consist of phosphatic pellets, bioclasts (bones), non-phosphatic minerals, and cement (Fig 3C and 3D). The isotropic phosphatic pellets occur as rounded to subrounded grains with dark and brown colours. These phosphatic pellets are partially replaced with cement materials such as calcedony and calcite. The anisotropic bioclasts are represented by angular to prismatic bone fragments and shells. The non-phosphatic materials are represented mainly by fine-grained quartz, mono- to crypto-crystalline chalcedony, calcite, and to a lesser extent, clay minerals. The cement/matrix is composed of cryptocrystalline calcareous and siliceous materials.

## 4. Results and discussion

### 4.1. Geochemical characteristics of the Hamadat phosphorites

Whole rock major oxides along with some trace elements concentrations of representative phosphorite samples are provided in Table 1. The $P_2O_5$ content shows a narrow range that

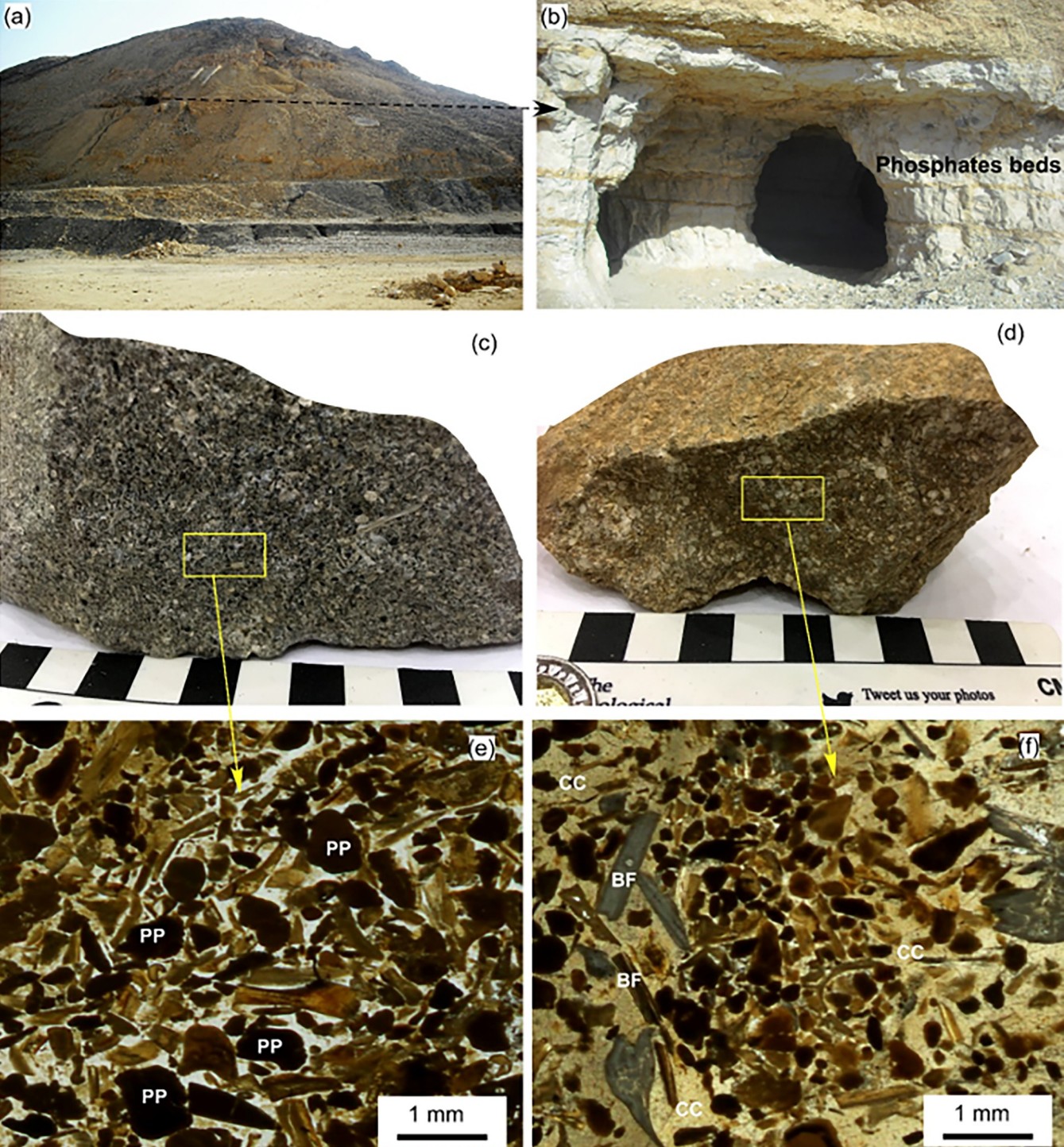

**Fig 2.** a) Field photographs showing the general overview of the Hamadat phosphatic mine, b) Close up photographs showing phosphatic beds from which the samples were collected, b-c) Various types of phosphate samples within the Hamadat Mine where c) shows the grey siliceous phosphorite and d) display the yellow calcareous phosphorite, e) photomicrographs show the rounded to subrounded phosphatic pellets (PP) surrounded by cryptocrystalline siliceous material, and f) bone fragments (BF), shells and phosphatic pellets impeded and surrounded with calcareous materials (CC).

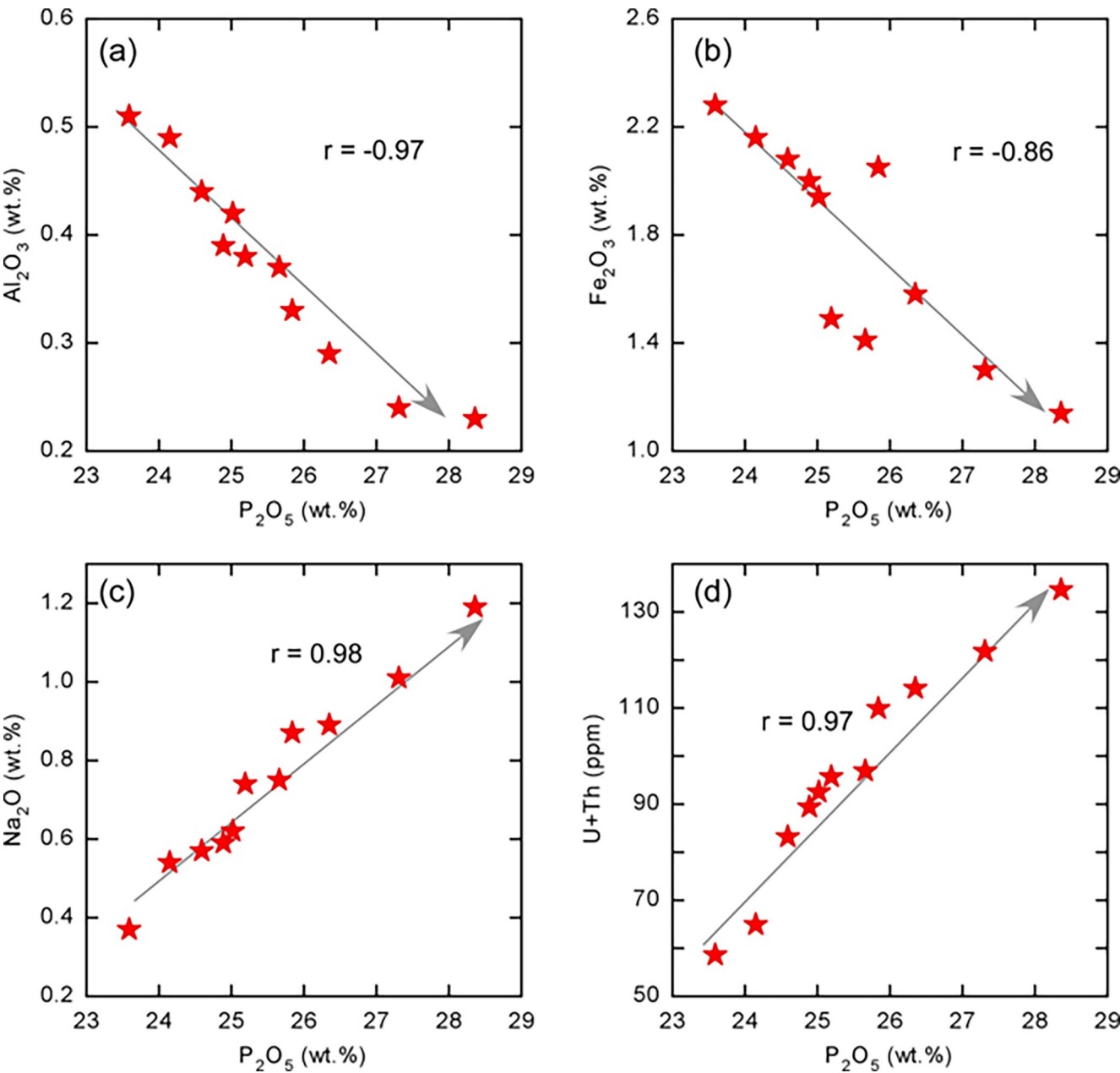

**Fig 3.** Binary relation between $P_2O_5$ and (a) $Al_2O_3$, (b) $Fe_2O_3$, (c) $Na_2O$, and (d) U + Th contents of the Hamadat phosphorites, where r is the calculated correlation coefficient.

varies from 23.59 wt.% to 28.36 wt.%, with an average of 25.54 wt.%. While $SiO_2$ shows a wide range that varies from 3.36 wt.% to 15.39 wt.%. The samples have a low concentration of $Al_2O_3$ (0.31–0.51 wt.%; avg. = 0.37±0.09 wt.%) and $TiO_2$ (0.01–0.05 wt.%; avg. = 0.03±0.01 wt.%). The $Al_2O_3$ concentration, which is the primary detrital component, has a typical inverse relationship with $P_2O_5$ (Fig 3A). Moreover, the samples have a low $Fe_2O_3$ (1.14–2.28 wt.%; avg. = 1.77±0.39 wt.%) content, which is also negatively correlated with the $P_2O_5$ (Fig 3B). The samples have a narrow range of CaO concentrations (40.85–44.35 wt.%; avg. = 41.83±1.09 wt.%) and low $Na_2O$ (0.37–1.19 wt.%; avg. = 0.74±0.24 wt.%) content, which is positively correlated

**Table 1. An example of a representative whole-rock geochemical study of the Hamadat Phosphorites that were analyzed (Central Eastern Desert, Egypt).**

| Sample | RP1 | RP2 | RP3 | RP4 | RP5 | RP6 | RP7 | RP8 | RP9 | RP10 | RP11 |
|---|---|---|---|---|---|---|---|---|---|---|---|
| $P_2O_5$ | 28.36 | 26.35 | 25.84 | 27.31 | 25.66 | 23.59 | 24.15 | 24.59 | 24.89 | 25.02 | 25.19 |
| $SiO_2$ | 3.36 | 4.25 | 4.49 | 4.71 | 4.86 | 15.39 | 15.26 | 14.19 | 12.58 | 10.84 | 9.92 |
| CaO | 41.21 | 42.04 | 40.94 | 41.81 | 40.85 | 44.35 | 41.25 | 42.91 | 40.97 | 41.19 | 42.61 |
| $Fe_2O_3$ | 1.14 | 1.58 | 2.05 | 1.3 | 1.41 | 2.28 | 2.16 | 2.08 | 2 | 1.94 | 1.49 |
| $Al_2O_3$ | 0.23 | 0.29 | 0.33 | 0.24 | 0.37 | 0.51 | 0.49 | 0.44 | 0.39 | 0.42 | 0.38 |
| $TiO_2$ | 0.02 | 0.03 | 0.02 | 0.03 | 0.02 | 0.01 | 0.03 | 0.02 | 0.03 | 0.05 | 0.03 |
| MnO | 0.18 | 0.15 | 0.09 | 0.13 | 0.18 | 0.13 | 0.14 | 0.18 | 0.11 | 0.13 | 0.08 |
| MgO | 1.39 | 0.98 | 1.22 | 0.94 | 0.87 | 1.46 | 0.69 | 0.57 | 0.74 | 0.42 | 0.37 |
| $Na_2O$ | 1.19 | 0.89 | 0.87 | 1.01 | 0.75 | 0.37 | 0.54 | 0.57 | 0.59 | 0.62 | 0.74 |
| $K_2O$ | 0.03 | 0.04 | 0.03 | 0.03 | 0.05 | 0.07 | 0.12 | 0.15 | 0.07 | 0.14 | 0.09 |
| Ba | 114 | 102 | 95 | 106 | 91 | 57 | 69 | 76 | 79 | 82 | 84 |
| Sr | 1316 | 1231 | 1219 | 1258 | 1219 | 896 | 948 | 998 | 1048 | 1101 | 1121 |
| Rb | 5 | 12 | 16 | 9 | 7 | 6 | 11 | 4 | 8 | 13 | 10 |
| Th | 6.6 | 5.1 | 4.9 | 5.8 | 4.9 | 3.6 | 3.9 | 4.2 | 4.4 | 4.5 | 4.7 |
| U | 128 | 109 | 105 | 116 | 92 | 55 | 61 | 79 | 85 | 88 | 91 |
| Zr | 28 | 21 | 19 | 22 | 31 | 29 | 35 | 27 | 24 | 28 | 22 |
| Cr | 79 | 54 | 67 | 62 | 59 | 48 | 56 | 74 | 101 | 92 | 117 |
| V | 125 | 103 | 101 | 119 | 99 | 56 | 60 | 65 | 72 | 78 | 80 |
| Ni | 45 | 39 | 28 | 51 | 40 | 29 | 35 | 24 | 19 | 33 | 21 |
| Zn | 259 | 194 | 304 | 98 | 405 | 169 | 239 | 309 | 157 | 270 | 253 |
| U/Th | 19.39 | 21.37 | 21.43 | 20.00 | 18.78 | 15.28 | 15.64 | 18.81 | 19.32 | 19.56 | 19.36 |
| Sr/Ba | 11.54 | 12.07 | 12.83 | 11.87 | 13.40 | 15.72 | 13.74 | 13.13 | 13.27 | 13.43 | 13.35 |

with $P_2O_5$ (Fig 3C). They also contain very low concentrations of $K_2O$ (0.03–0.15 wt.%; avg. = 0.74±0.04 wt.%) and MnO (0.08–0.18 wt.%; avg. = 0.14±0.03 wt.%), suggesting the low contribution of the detrital material during the deposition of the studied phosphates.

The samples contain high U concentration range from 55 to 128 ppm, with an average of 92 ±22 ppm, which suggest that they probably are riniferous phosphorites formed at deep-water levels [31]. It is important to note that the U concentrations in the Hamadat phosphorites is quite similar to those recorded in Tébessa (Algeria) phosphorites (U = up to 126 ppm) [32,33], but are quite higher than those recorded in Metlaoui (Tunisia) phosphorites (average U > 50 ppm) [34,35]. These high U concentrations are higher than those of the UCC (2.5 ppm; [36] and the PAAS 2.7 ppm; [37]. The samples contain high Sr (avg. = 1123±138 ppm), Ba (avg. = 87±17 ppm), Cr (avg. = 74±22 ppm), V (avg. = 87±24 ppm), and Zn (avg. = 242±85 ppm) concentrations, which could be due to biologic enrichment [38]. Th, Zr, and Rb occur at deficient concentrations (Table 1), which strongly supports the low detrital input during forming of the Hamadat phosphorites. The value of the U/Th ratio is high (15.28–21.43) and the content of U and Th is positively correlated with $P_2O_5$ (Fig 3D), which could be attributed to the substitution of U for Ca in the apatite structure. The high Sr/Ba values (12–16) of the studied phosphorites indicate that they were generated by normal marine sedimentation [38]. The wide occurrence of phosphatic components (e.g., apatite), that accommodate U with high concentrations is the main reason for the high Radioactivity of the studied phosphorites.

## 4.2. Activity concentrations and radiological parameters

Radioactive elements such as uranium phosphorite are available in phosphate rock. The presence of phosphate components that contain uranium is the leading cause of the radioactive

**Table 2. Radioelement activity concentrations (A) ($^{226}$Ra, $^{232}$Th, and $^{40}$K).**

| Region Name | Samples | A (Bq kg$^{-1}$) $^{226}$Ra | A (Bq kg$^{-1}$) $^{232}$Th | A (Bq kg-1) $^{40}$K |
|---|---|---|---|---|
| Hamadat mine area | Rb1 | 286±14.3 | 127±6.4 | 68±3.4 |
| | Rb2 | 174±8.7 | 158±7.9 | 30±1.5 |
| | Rb 3 | 289±14.4 | 116±5.8 | 123±6.1 |
| | Rb 4 | 37.0±1.8 | 7.00±0.3 | 49±2.5 |
| | Rb 5 | 128±6.4 | 140±7.0 | 56±2.8 |
| | Rb 6 | 263±13.1 | 117±5.8 | 63±3.1 |
| | Rb 7 | 103±5.2 | 113±5.7 | 42±2.1 |
| | Rb 8 | 127±6.3 | 139±6.9 | 55±2.8 |
| | Rb 9 | 173±8.6 | 189±9.5 | 75±3.8 |
| | Rb 10 | 251±12.5 | 101±5.0 | 107±5.3 |
| | Rb 11 | 195±9.8 | 177±8.9 | 34±1.7 |
| **Average** | | **184.18±9.19** | **125.82±6.29** | **63.82±3.19** |

abnormalities within and/or of their apatite minerals [39]. The $^{226}$Ra, $^{232}$Th, and $^{40}$K concentrations in phosphate samples from the Hamadat mine area in units of Bq kg$^{-1}$ are presented in Table 2. From Table 2, the activity concentration (Bq kg$^{-1}$) of $^{226}$Ra ranged from 37.0±1.8 in the RP4 sample to 289±14.4 in the RP3 sample, with an average value of 184.18±9.19. For $^{232}$Th the activity ranged from 7.00±0.3 in a sample coded by RP4 to 189±9.5 Bqkg$^{-1}$ in a sample coded by RP9 with an average value of 125.82±6.29. Finally, the activity of $^{40}$K ranged from 30±1.5 in the sample code by RP2 to 123±6.1 in the sample code by RP3, with an average value of 63.82±3.19 Bqkg$^{-1}$. Phosphate sample of RP3 presented the highest activity for $^{226}$Ra and $^{40}$K, while phosphate sample of RP9 presented the highest activity for $^{232}$Th. The lowest activities for $^{226}$Ra, $^{232}$Th and $^{40}$K have been recorded in samples coded by Rb4 as shown in Fig 4.

According to the worldwide average concentrations (35, 35, and 400 Bqkg-1) for $^{226}$Ra, $^{232}$Th, and $^{40}$K, respectively, which were reported by [40], our findings demonstrate that the samples' average value of $^{40}$K under investigation is lower than the world average, while the average value of $^{226}$Ra and $^{232}$Th are higher than the world average concentrations [2,41–44]. The variations in the activity concentration are associated with the methods by which the soils are concentrated as well as the radioactive distribution in the rocks from which they arise. The activity concentration of $^{226}$Ra, $^{232}$Th, and $^{40}$K (Bq kg-1) in phosphate rocks compared to the data of similar studies from different countries are presented in Table 3.

Table 4 lists the values for the following parameters: radium equivalent activity (Ra$_{eq}$), dose rate (D), exterior and internal hazards indices (H$_e$x, H$_{in}$), annual effective dose (AED), and excess lifetime cancer risk (ELCR). Ra$_{eq}$ values for the phosphate samples ranged from 80.8 to 474 Bq kg$^{-1}$, with a mean value of 371.7 Bq kg-1, as shown in Table 3. These values are greater than the advised limit of 370 Bq kg$^{-1}$ [23,49].

The dose rate values ranged from 22.6 to 207.7, with a mean value of 159.9 nGyh$^{-1}$, which is higher than the international limit of 59 nGyh$^{-1}$ [50]. As shown in columns 5 and 6 of Table 4, the mean values of external and internal hazards for phosphate samples are higher than the unity (permissible level) [51–54], which causes harm to the populations in the region under investigation. On the other hand, the yearly effective dose values were more significant than the global average effective dose, about 70 μSvy$^{-1}$, and varied from 27.718 to 254.708 Svy$^{-1}$ with a mean value of 196.1 Svy$^{-1}$ [22]. Finally, the excess lifetime cancer risk values were lower than the global limit of 29 10–3, ranging from 9.70 E-05 to 8.91E-04.

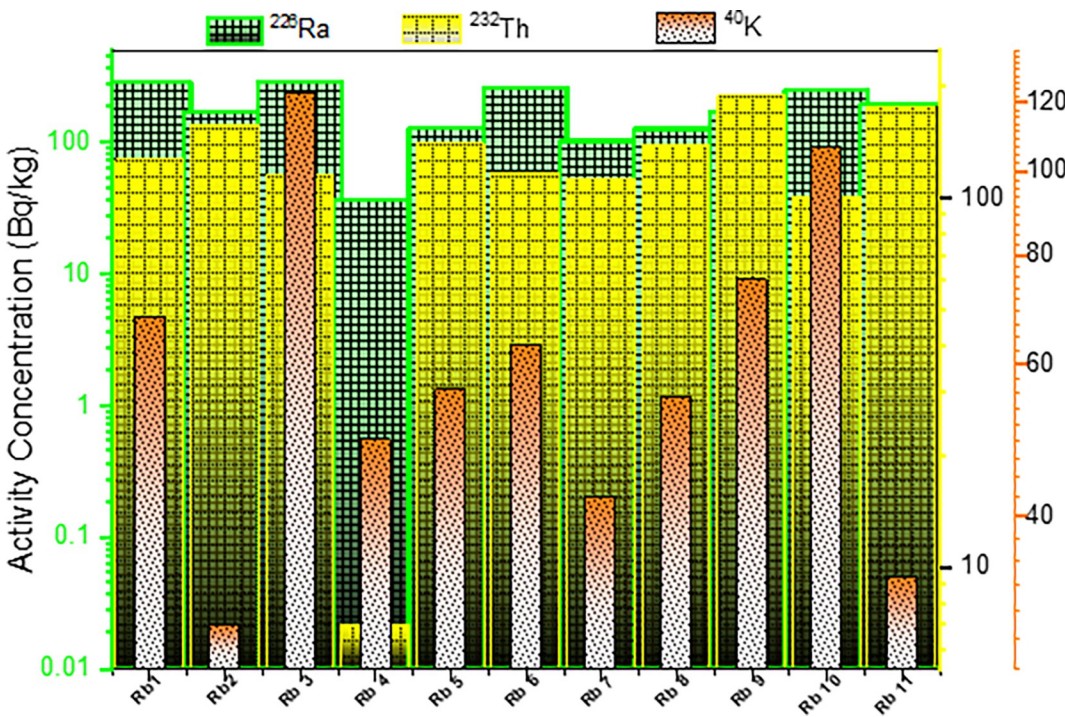

**Fig 4. $^{226}$Ra, $^{232}$Th, and $^{40}$K Radioelement activity concentrations.**

## 4.3. Statistical analysis

Table 5 displays the statistical characteristics of the data that were collected (including the lowest, median, range, mean, maximum, standard deviation, standard error, skewness, and kurtosis of the radionuclides for each of the phosphate samples). There is no correspondence between the mean activity concentrations as shown in Table 5. The value of $^{40}$K has a positive skewness, which indicates that the distribution it represents has an unbalanced tail that extends towards more positive values. On the other hand, a negative skewness indicates a distribution with an asymmetric tail that extends towards greater negative values, which is exactly what we see in the case of $^{226}$Ra and $^{232}$Th. In conclusion, $^{232}$Th and $^{40}$K both show a positive kurtosis, indicating their distributions are relatively peaky [41,45].

**Table 3. Phosphate rocks from a variety of countries were tested to determine the activity concentration of $^{226}$Ra, $^{232}$Th and $^{40}$K (Bq kg$^{-1}$).**

| Country | A (Bq kg$^{-1}$) $^{226}$Ra | A (Bq kg$^{-1}$) $^{232}$Th | A (Bq kg-1) $^{40}$K | References |
|---|---|---|---|---|
| Egypt (Abu-Tartur) | 117.6 | 65 | 126 | [45] |
| Egypt (El-Mahamid) | 567 | 217.3 | 217.3 | [46] |
| Egypt (W. El-Mashash) | 666 | 329.4 | 329.4 | [46] |
| Egypt (El-Sibaiya) | 538 | 2.5 | N.F | [47] |
| Egypt (El-Quseir) | 358 | 38 | N.F | [47] |
| Egypt (Abu-Zaabal) | 214 | 37 | 19 | [48] |
| Finland | 10 | 10 | 110 | [38] |
| Pakistan (Hazara) | 440 | 50 | 207 | [38] |
| Tanzania (Arusha) | 5022 | 717 | 286 | [46] |
| **Egypt(the Hamadat mine area)** | **184.18±9.19** | **125.82±6.29** | **63.82±3.19** | **This work** |

**Table 4. Annual effective dose (AED), radium equivalent activity (Ra$_{eq}$), dose rate (D), external hazards index (H$_{ex}$) and internal hazards index (Hin), and excess lifetime cancer risk (ELCR) were measured for the samples.**

| Region Name | Samples | Ra$_{eq}$ (BqKg$^{-1}$) | D (nGyh$^{-1}$) | H$_{ex}$ | H$_{in}$ | AED (μSvy$^{-1}$) | ELCR |
|---|---|---|---|---|---|---|---|
| Hamadat mine area | Rb1 | 474.0 | 207.7 | 1.281 | 2.055 | 254.708 | 8.91E-04 |
| | Rb2 | 401.4 | 172.3 | 1.084 | 1.554 | 211.300 | 7.40E-04 |
| | Rb 3 | 464.4 | 204.5 | 1.255 | 2.035 | 250.798 | 8.78E-04 |
| | Rb 4 | 80.8 | 22.6 | 0.135 | 0.233 | 27.718 | 9.70E-05 |
| | Rb 5 | 332.8 | 142.4 | 0.899 | 1.245 | 174.694 | 6.11E-04 |
| | Rb 6 | 434.5 | 190.4 | 1.774 | 1.884 | 233.483 | 8.17E-04 |
| | Rb 7 | 268.5 | 114.9 | 0.725 | 1.004 | 140.901 | 4.93E-04 |
| | Rb 8 | 329.1 | 140.9 | 0.889 | 1.231 | 172.774 | 6.05E-04 |
| | Rb 9 | 449.2 | 192.3 | 1.213 | 1.680 | 235.836 | 8.25E-04 |
| | Rb 10 | 403.1 | 177.5 | 1.089 | 1.766 | 217.706 | 7.62E-04 |
| | Rb 11 | 451.3 | 193.7 | 1.219 | 1.747 | 237.599 | 8.32E-04 |
| **Average** | | **371.7** | **159.9** | **1.1** | **1.5** | **196.1** | **6.86E-04** |

Fig 5 shows the correlation coefficients between the radionuclides $^{226}$Ra, $^{232}$Th, and $^{40}$K and radiation hazard parameters at the Hamadat mine area. From Fig 5 (A1, A2 and A3), we can see that $^{226}$Ra showed strong positive correlations with hazard parameters (ELCR, AEDE, and D), while $^{40}$K showed weak Correlation, as shown in Fig 5 (B1, B2, and B3). This indicates that the $^{226}$Ra was the essential reason for the hazard parameters. A comparison of natural radioactivity studies in different countries is shown in Table 6. The natural radioactivity concentrations of different regions clearly vary.

In this particular study on the Hamadat phosphorites, the geochemical characteristics of the samples were analyzed, including their major oxides and trace element concentrations. The P$_2$O$_5$ content was found to have a narrow range, while SiO$_2$ showed a wide range. The samples had low concentrations of Al$_2$O$_3$ and TiO$_2$, which were negatively correlated with P2O5. The samples also had low Fe$_2$O$_3$ content, which was also negatively correlated with P$_2$O$_5$, and a narrow range of CaO concentrations. They contained very low concentrations of K$_2$O and MnO, which suggested low detrital material contribution during the deposition of the studied phosphates. The samples were also found to contain high concentrations of U, Sr,

**Table 5. Descriptive statistics of $^{238}$U, $^{232}$Th and $^{40}$K (Bq kg$^{-1}$) in the Hamadat mine area.**

| | $^{226}$Ra | $^{232}$Th | $^{40}$K |
|---|---|---|---|
| Number of values | 11 | 11 | 11 |
| Minimum | 37 | 7 | 30 |
| 25% Percentile | 127 | 113 | 42 |
| Median | 174 | 127 | 56 |
| 75% Percentile | 263 | 158 | 75 |
| Maximum | 289 | 189 | 123 |
| Range | 252 | 182 | 93 |
| Mean | 184.2 | 125.8 | 63.82 |
| Standard Deviation | 81.89 | 48.06 | 28.94 |
| Standard Error of Mean | 24.69 | 14.49 | 8.727 |
| Skewness | -0.2434 | -1.4 | 1.055 |
| Kurtosis | -0.8229 | 3.521 | 0.5727 |

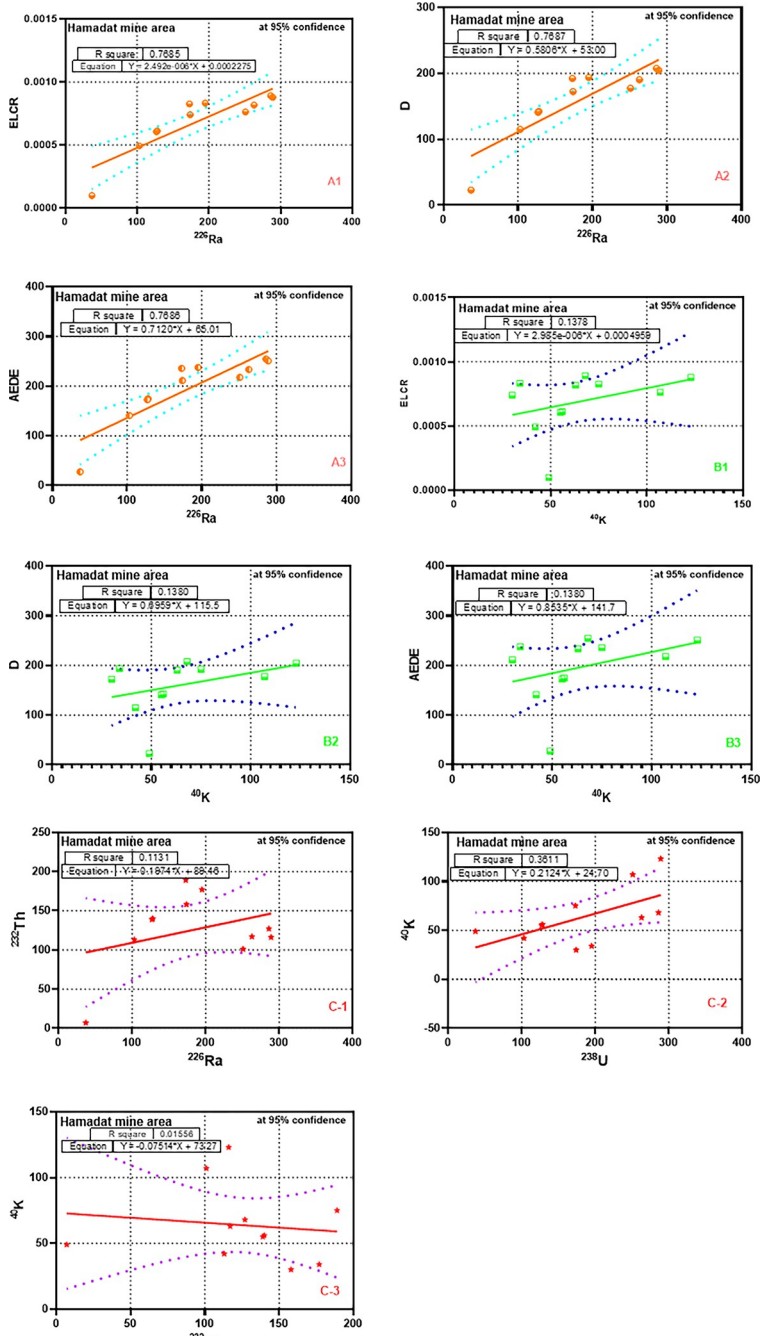

**Fig 5. Correlation coefficients illustrate the relation between the radionuclides $^{226}$Ra, $^{232}$Th, $^{40}$K, and radiation hazard parameters at the Hamadat mine area.**

Ba, Cr, V, and Zn, which could be due to biologic enrichment. Th, Zr, and Rb occurred at deficient concentrations, which supported the low detrital input during the formation of the phosphorites. The U/Th ratio was high, and the content of U and Th was positively correlated with $P_2O_5$, which could be attributed to the substitution of U for Ca in the apatite structure. The high Sr/Ba values indicated that the phosphorites were generated by normal marine sedimentation.

**Table 6. Activity concentrations of $^{226}$Ra, $^{238}$U, $^{232}$Th and $^{40}$K in phosphate deposits from different countries.**

| Countries | Activity concentration (Bq / kg1) | | | | |
|---|---|---|---|---|---|
| | $^{40}$K | $^{232}$Th | $^{238}$U | $^{226}$Ra | References |
| Algeria | 81.29 | 65.53 | 788.34 | 570.47 | [55] |
| China | - | 25 | 150 | 150 | [56] |
| Egypt | 822.76 | 131.26 | - | 215.43 | [57] |
| Finland | 110 | 10 | - | 10 | [56] |
| Russia | 40 | 80 | 40 | 30 | [56] |
| Morocco | 10 | 30 | 1600 | 1700 | [58] |
| Nigeria | 40 | 20 | - | 558 | [59] |
| Pakistan | 206 | 52 | 550 | 511 | [56] |
| Saudi Arabia | 250 | 40 | | 519 | [60] |
| Tunisia | 32 | 29 | 580 | 821 | [56] |
| turkey | 256 | 26 | 557 | 625 | [61] |
| Western Sahara | 30 | 7 | 900 | 900 | [56] |
| Egypt | 63.82 | 125.82 | - | 184.18 | Present study |

The activity concentrations and radiological parameters of the samples were also analyzed. The $^{226}$Ra, $^{232}$Th, and $^{40}$K concentrations in phosphate samples from the Hamadat mine area were presented, and the average values for $^{226}$Ra and $^{232}$Th were found to be higher than the world average concentrations. The average value for $^{40}$K was found to be lower than the world average. The variations in the activity concentration were associated with the methods by which the soils were concentrated as well as the radioactive distribution in the rocks from which they arise. The radium equivalent activity, dose rate, exterior and internal hazards indices, annual effective dose, and excess lifetime cancer risk were also calculated and presented. Overall, the results and discussion section provides a detailed analysis of the findings obtained from the research conducted on the Hamadat phosphorites.

## 5. Conclusions

The phosphorite samples were collected from the Duwi Formation late Cretaceous horizontal strata at the Hamadat phosphatic mine, which is located in the Central Eastern Desert of Egypt. The samples consist of bioclasts and crypto-crystalline apatite (collophane), which are partially replaced with cement materials (calcedony and calcite). The samples contain high $P_2O_5$, CaO, U, Sr, Ba, Cr, V and Zn concentrations, with a notably very low content of $Al_2O_3$, $Fe_2O_3$, $Na_2O$, $K_2O$, MnO, Th, Zr and Rb. The high Sr/Ba values indicate that phosphorites were generated by normal marine sedimentation. The high U/Th ratio could be attributed to the substitution of U for Ca in the apatite structure, which also increases the Radioactivity within the studied phosphorites. The spectrometer of Gamma-ray includes a high-scintillation detector of resolution NaI (Tl) crystal 3×3 has been used to measure occurring radionuclides in the studied phosphorites. The levels of natural Radioactivity in the samples are more significant than permitted. Additionally, the mean radiological risk values were higher than the world average, making it unsafe for the population living there. The information gathered here serves as a reference point for creating an environmental map of the Hamadat phosphate mine area while also showing that any potential effects of the mining operations if any exist, are minimal and obscured by variations in the background values that are inherent to nature.

## Author Contributions

**Conceptualization:** Douaa Fathy, Hesham M. H. Zakaly, Saad S. Alarifi, Antoaneta Ene.

**Data curation:** Mabrouk Sami.

**Formal analysis:** Saad S. Alarifi.

**Funding acquisition:** Antoaneta Ene.

**Investigation:** El Saeed R. Lasheen, Mabrouk Sami.

**Methodology:** El Saeed R. Lasheen, Reda Elsaman, Mabrouk Sami.

**Software:** Hesham M. H. Zakaly, Reda Elsaman, Antoaneta Ene.

**Supervision:** Hesham M. H. Zakaly, Antoaneta Ene.

**Validation:** Hamdy A. Awad.

**Visualization:** Reda Elsaman, Hamdy A. Awad.

**Writing – original draft:** Douaa Fathy, Hesham M. H. Zakaly, El Saeed R. Lasheen.

**Writing – review & editing:** Hesham M. H. Zakaly, Reda Elsaman, Saad S. Alarifi, Hamdy A. Awad, Antoaneta Ene.

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
