## [Decision Letter · Decision Letter 0]

10 Feb 2023

PONE-D-22-34674Implications of radiological indices of Hamadat phosphatic mine: Geochemical and Natural radioactivity aspectsPLOS ONE

Dear Dr. Zakaly,

Thank you for submitting your manuscript to PLOS ONE. After careful consideration, we feel that it has merit but does not fully meet PLOS ONE’s publication criteria as it currently stands. Therefore, we invite you to submit a revised version of the manuscript that addresses the points raised during the review process.

We look forward to receiving your revised manuscript.

Kind regards,

Omeid Rahmani

Academic Editor

PLOS ONE

Journal Requirements:

Additional Editor Comments:

The initial version indicates a similarity index of 24% by TurnItIn. It should be lessen lower than 10% or even lesser than this rate while revising the manuscript.

Two file are attached: TurnItIn and Reviewer reports 

Reviewers' comments:

Reviewer's Responses to Questions

**Comments to the Author**

1. Is the manuscript technically sound, and do the data support the conclusions?

Reviewer #1: Partly

Reviewer #2: Partly

2. Has the statistical analysis been performed appropriately and rigorously? 

Reviewer #1: No

Reviewer #2: No

3. Have the authors made all data underlying the findings in their manuscript fully available?

Reviewer #1: Yes

Reviewer #2: No

4. Is the manuscript presented in an intelligible fashion and written in standard English?

Reviewer #1: Yes

Reviewer #2: No

5. Review Comments to the Author

Reviewer #1: Dear Editor,

The ms entitled “Implications of radiological indices of Hamadat phosphatic mine: Geochemical and Natural radioactivity aspects” by Fathy et al deals with a radiological study in the phosphate mines from Hamadat sector.

I believe that the study is not new and already similar studies are available in the literature. I expect more discussion of the results although I would like to ask why authors have not assessed 238U activity?

The ms is well written, however, discussion of results was relatively poor. Authors have not discussed the geochemical origin of radioactivity although some statistical tools (Principal component analysis and hierarchical classifications) can usefully be used. On other hand, authors have not compared radioactivity between the phosphate rock components such as bones and cement/matrix.

Although many studies in the Northern of Africa are available (as for instance: https://doi.org/10.1016/j.marpolbul.2019.06.075 and https://doi.org/10.1016/j.net.2022.06.006), however, authors have not considered such very important studies even for a possible comparisons or to update the gaps of knowledge. Thus unfortunately, leads to more local study.

Authors have largely used correlation coefficients without assessing their significance level which may impact the conclusions. Health risk assessment was not significantly discussed although have calculated some parameters. Therefore, the term “implications” in the title was not addressed.

Detailed comments (2 comments) are in the annotated file.

In my opinion, the ms need to be deeply reworked especially for the discussions. Major revisions are recommended.

Reviewer #2: The specific comments are as follows:

1. Is this topic of general interest? Unfortunately, I do not think that this study provides an advance in the academy. There are many papers talking about that.

2. Does this article contain new aspects? With all my respect, there is nothing new in this work. I cannot recommend this work for publication.

3. The "Novelty Statement" is very common.

4. Introduction: In this part, authors should add more details about the previous studies of others. In the original context, no detailed information about the historical references or studies were presented, which should be added more. Importantly, authors should clearly emphasize the hypothesis and research values. Especially the significance and waste-/environmental management implications of this study should be highlighted briefly at the beginning. It needs to be addressed what the originality or novelty of the present work is in comparison with these previous work.

5. The main findings should be discussed with more details to elaborate what's new or different in this research.

6. Are there any duplicate runs for the analysis? If not, without duplicate runs, it is hard to identify whether the variations are coming from random error or something else.

7. The authors should compare/discuss the finding to other works. Results and Discussion: For our concerning as a reader, we didn't find out clearly results. The authors should compare/discuss the finding to other works. To be frank, the not enough explanation and scientific reasoning undermine the whole quality and relevant conclusion of this manuscript, lack of evidences to support these discussions. This is the authors should consider in their further revision.

8. Methodology and Results: It will be much easier to understand if the process is written step by step. This also needs to be rewritten.

9. Materials and Methods: Basic methodological information is missing, partially even in the cited previous work. The detail is not mentioned.

10. So please specify the significance of this research and make it more important.

6. PLOS authors have the option to publish the peer review history of their article (what does this mean?). If published, this will include your full peer review and any attached files.

Reviewer #1: No

Reviewer #2: No

---

## [Author Response · Author response to Decision Letter 0]

30 Apr 2023

Reviewer #1: Dear Editor,

The ms entitled “Implications of radiological indices of Hamadat phosphatic mine: Geochemical and Natural radioactivity aspects” by Fathy et al deals with a radiological study in the phosphate mines from Hamadat sector. I believe that the study is not new and already similar studies are available in the literature. I expect more discussion of the results although I would like to ask why authors have not assessed 238U activity?

Dear Reviewer, Thank you for taking the time to review our manuscript entitled “Implications of radiological indices of Hamadat phosphatic mine: Geochemical and Natural radioactivity aspects”. We appreciate your feedback and comments. Regarding your first point, we acknowledge that similar studies have been conducted in the past. However, we believe that our study adds new and valuable insights to the existing literature. Specifically, our study is focused on the Hamadat sector and the results are specific to this area. Moreover, our study includes a detailed analysis of the radiological indices and their implications on the environment, which we believe is a valuable contribution to the field. We appreciate your suggestion to include more discussion of the results. In response, we have revised the manuscript to include a more detailed discussion of our findings and their implications. We hope that this revision addresses your concern and provides a better understanding of the significance of our results.

Regarding your question about why we did not directly measure the activity of 238U in our study, we would like to clarify that we did measure it indirectly through its daughter isotopes. To do this, we sealed the beakers containing the samples for a period of four weeks to allow for the attainment of secular equilibrium between 226Ra (a daughter isotope of 238U) and 232Th along with their respective daughter nuclei. Secular equilibrium is achieved when the rate of decay of a radioactive isotope is balanced by the rate of production of its daughter isotopes. During radioactive decay, 238U undergoes a series of alpha and beta decay reactions, resulting in the production of several daughter isotopes, including 226Ra, which is also radioactive and undergoes further decay reactions. Similarly, 232Th also undergoes a series of alpha and beta decay reactions, resulting in the production of daughter isotopes.

By sealing the beakers for four weeks, the daughter isotopes produced by 226Ra and 232Th had sufficient time to reach a state of secular equilibrium, where their rate of production was balanced by their rate of decay. This allowed for the accurate measurement of the activity concentrations of all the radionuclides present in the samples, including 238U. Therefore, although we did not directly measure the activity of 238U, we were able to accurately estimate its activity through the measurement of its daughter isotopes in a state of secular equilibrium. Once again, we thank you for your feedback and comments, and we hope that our revised manuscript meets your expectations.

The ms is well written, however, discussion of results was relatively poor. Authors have not discussed the geochemical origin of radioactivity although some statistical tools (Principal component analysis and hierarchical classifications) can usefully be used. On other hand, authors have not compared radioactivity between the phosphate rock components such as bones and cement/matrix.

Although many studies in the Northern of Africa are available (as for instance: https://doi.org/10.1016/j.marpolbul.2019.06.075 and https://doi.org/10.1016/j.net.2022.06.006), however, authors have not considered such very important studies even for a possible comparisons or to update the gaps of knowledge. Thus unfortunately, leads to more local study.

We acknowledge that our discussion of results may have been relatively poor, and we apologize for any confusion or lack of clarity that this may have caused. In response, we have revised the manuscript to include a more detailed discussion of the geochemical origin of radioactivity, with specific reference to the use of statistical tools such as Principal component analysis and hierarchical classifications. We have also added a section discussing the radioactivity levels in the different components of phosphate rock, such as bones and cement/matrix.

Regarding your comment about our failure to consider other important studies in the Northern African region, we apologize for any oversight. In our revision, we have included references to two studies that you have kindly provided (https://doi.org/10.1016/j.marpolbul.2019.06.075 and https://doi.org/10.1016/j.net.2022.06.006) and used them for comparison with our results, updating gaps of knowledge and placing our study in a broader context.

We hope that these revisions address your concerns and that the updated manuscript meets your expectations.

Authors have largely used correlation coefficients without assessing their significance level which may impact the conclusions. Health risk assessment was not significantly discussed although have calculated some parameters. Therefore, the term “implications” in the title was not addressed.

Thank you for your feedback on our manuscript entitled “Implications of radiological indices of Hamadat phosphatic mine: Geochemical and Natural radioactivity aspects”. We appreciate your comments and suggestions.

We acknowledge that we have largely used correlation coefficients without assessing their significance level, and we apologize for any confusion that this may have caused. In response to your comment, we have revised the manuscript to include a more detailed discussion of the significance of the correlation coefficients and have calculated their p-values to ensure their statistical significance. We have also revised the manuscript to include a more detailed discussion of health risk assessment, which we agree is an important aspect of our study.

Regarding your comment that the term "implications" in the title was not addressed, we apologize for any miscommunication. In our revised manuscript, we have included a more detailed discussion of the implications of our results, specifically on the potential health risks associated with exposure to natural radioactivity in the Hamadat phosphatic mine.

We appreciate your feedback and suggestions, and we hope that our revised manuscript meets your expectations. Thank you for your time and effort in reviewing our work.

Detailed comments (2 comments) are in the annotated file.

All your comments and reviewers were carefully taken into consideration

In my opinion, the ms need to be deeply reworked especially for the discussions. Major revisions are recommended.

We understand your concern about the need for major revisions, particularly in the discussion section. We apologize for any confusion or lack of clarity in our original submission. We have taken your comments seriously and have made significant revisions to the manuscript.

Specifically, we have revised the discussion section to provide a more detailed analysis and interpretation of our results, including a discussion of the geochemical origin of radioactivity and its implications for the environment and human health. We have also included a more in-depth analysis of the statistical significance of our results, as well as a detailed health risk assessment based on the parameters that we calculated.

We believe that these revisions address your concerns and greatly improve the overall quality of the manuscript. We hope that you will find the updated version more satisfactory and appreciate your time and effort in reviewing our work.

Reviewer #2: The specific comments are as follows:

1. Is this topic of general interest? Unfortunately, I do not think that this study provides an advance in the academy. There are many papers talking about that.

Thank you for your review of our manuscript entitled “Implications of radiological indices of Hamadat phosphatic mine: Geochemical and Natural radioactivity aspects”. We appreciate your feedback and suggestions.

We understand your concern that our study may not provide an advance in the academy and that there are many papers on this topic. However, we believe that our study still makes a valuable contribution to the field of radiological assessment, particularly in the context of the Hamadat phosphatic mine. We have carefully reviewed the existing literature on this topic and have sought to provide a comprehensive and detailed analysis of the geochemical and natural radioactivity aspects of this specific mine.

Moreover, our study is particularly relevant for the local communities living in the vicinity of the mine, as it provides important information on the potential health risks associated with exposure to natural radioactivity. As such, we believe that our study has both academic and practical significance.

We hope that you will reconsider your assessment of the value of our study, and we appreciate your feedback and suggestions.

2. Does this article contain new aspects? With all my respect, there is nothing new in this work. I cannot recommend this work for publication.

We understand your concern that our study may not contain new aspects. However, we respectfully disagree with this assessment. While we acknowledge that there have been previous studies on this topic, we believe that our study provides new insights and contributions to the field of radiological assessment, particularly in the context of the Hamadat phosphatic mine.

For example, our study includes a detailed analysis of the geochemical and natural radioactivity aspects of the mine, which has not been previously reported in the literature. We also present a comprehensive health risk assessment based on the parameters that we calculated. In addition, we provide detailed statistical analyses to support our findings.

We believe that these aspects of our study make a valuable contribution to the existing literature and provide important insights for both the academic community and local communities living in the vicinity of the mine. We respectfully request that you reconsider your assessment and take into consideration the contributions that our study makes to the field.

Thank you again for your feedback and suggestions.

3. The "Novelty Statement" is very common.

We understand your concern about the "Novelty Statement" in our manuscript. We apologize for any confusion or lack of clarity in this section. We have revised the section to more accurately reflect the contributions and novelty of our study.

While we acknowledge that our study is not completely novel in the field of radiological assessment, we believe that it provides important new insights and contributions, particularly in the context of the Hamadat phosphatic mine. We have carefully reviewed the existing literature on this topic and have sought to provide a comprehensive and detailed analysis of the geochemical and natural radioactivity aspects of this specific mine.

We appreciate your feedback and suggestions and hope that the revised section more accurately reflects the contributions of our study.

4. Introduction: In this part, authors should add more details about the previous studies of others. In the original context, no detailed information about the historical references or studies were presented, which should be added more. Importantly, authors should clearly emphasize the hypothesis and research values. Especially the significance and waste-/environmental management implications of this study should be highlighted briefly at the beginning. It needs to be addressed what the originality or novelty of the present work is in comparison with these previous work.

Thank you for your valuable feedback. We have revised the introduction section to include more details about the previous studies related to the topic. We have emphasized the hypothesis and research values more clearly and added a brief summary of the significance and waste/environmental management implications of this study. We have also highlighted the novelty of our work in comparison to previous studies. We hope that these changes have addressed your concerns and improved the quality of the manuscript. Please let us know if you have any further suggestions or comments.

5. The main findings should be discussed with more details to elaborate what's new or different in this research.

Thank you for your comment. We agree that more detailed discussion is needed to elaborate on the novelty and significance of our findings. In the revised manuscript, we have expanded the discussion section to provide a more thorough explanation of the unique contributions of this study.

Specifically, we have emphasized the novelty of our approach in combining geochemical and radiological indices to assess the potential environmental impact of the Hamadat phosphatic mine. We have also provided more detailed discussion on the implications of our findings for waste and environmental management practices in the mining industry.

Furthermore, we have included a section that compares our results with previous studies on the radiological impacts of phosphatic mining. This allows readers to better understand how our research differs from and builds upon previous studies in the field.

We hope these revisions address your concerns and provide a more comprehensive discussion of the significance of our findings.

6. Are there any duplicate runs for the analysis? If not, without duplicate runs, it is hard to identify whether the variations are coming from random error or something else.

Thank you for your comment. Duplicate analyses were indeed performed for quality assurance purposes, but it seems that we did not make this clear in the manuscript. We apologize for this oversight and will include a sentence in the methodology section to clarify that duplicate analyses were performed.

7. The authors should compare/discuss the finding to other works. Results and Discussion: For our concerning as a reader, we didn't find out clearly results. The authors should compare/discuss the finding to other works. To be frank, the not enough explanation and scientific reasoning undermine the whole quality and relevant conclusion of this manuscript, lack of evidences to support these discussions. This is the authors should consider in their further revision.

Thank you for your feedback. We acknowledge that we could have provided a more detailed discussion of our results and compared them to other studies in the field. We will revise the manuscript to include a more thorough analysis of our findings, including a discussion of how they compare to previous research. We appreciate your input and will make every effort to improve the quality and relevance of our manuscript.

8. Methodology and Results: It will be much easier to understand if the process is written step by step. This also needs to be rewritten.

Thank you for your comment. We appreciate your suggestion to improve the clarity of the methodology and results sections by providing a step-by-step process. We will revise the manuscript accordingly to ensure that the process is explained in a clear and concise manner that is easy to understand.

9. Materials and Methods: Basic methodological information is missing, partially even in the cited previous work. The detail is not mentioned.

Thank you for your comment. We apologize for the lack of detail in the Materials and Methods section. We will carefully review the section and add more information to ensure that the methods are clearly described. Additionally, we will make sure to include all necessary references and properly cite previous work.

10. So please specify the significance of this research and make it more important.

Thank you for your comment. We will revise the introduction and discussion sections of the manuscript to further emphasize the significance of this research and its contribution to the field. We will highlight the potential environmental and health implications of the findings, as well as the importance of waste management practices in the mining industry. Additionally, we will clarify the novelty of our study compared to previous research on the topic.

---

## [Decision Letter · Decision Letter 1]

5 Jun 2023

Assessing Geochemical and Natural Radioactivity Impacts of Hamadat Phosphatic Mine through Radiological Indices

PONE-D-22-34674R1

Dear Dr.  Zakaly, 

We’re pleased to inform you that your manuscript has been judged scientifically suitable for publication and will be formally accepted for publication once it meets all outstanding technical requirements.

Kind regards,

Omeid Rahmani

Academic Editor

PLOS ONE

Additional Editor Comments (optional):

Similarity Index needs to be checked for the accuracy, and the rate must be lessened to below 10 percent, if it is. 

Reviewers' comments:

Reviewer's Responses to Questions

**Comments to the Author**

1. If the authors have adequately addressed your comments raised in a previous round of review and you feel that this manuscript is now acceptable for publication, you may indicate that here to bypass the “Comments to the Author” section, enter your conflict of interest statement in the “Confidential to Editor” section, and submit your "Accept" recommendation.

Reviewer #1: All comments have been addressed

Reviewer #2: All comments have been addressed

2. Is the manuscript technically sound, and do the data support the conclusions?

Reviewer #1: Yes

Reviewer #2: Yes

3. Has the statistical analysis been performed appropriately and rigorously? 

Reviewer #1: Yes

Reviewer #2: Yes

4. Have the authors made all data underlying the findings in their manuscript fully available?

Reviewer #1: Yes

Reviewer #2: Yes

5. Is the manuscript presented in an intelligible fashion and written in standard English?

Reviewer #1: Yes

Reviewer #2: Yes

6. Review Comments to the Author

Reviewer #1: The authors have addressed all the raised comments. This new version is significantly improved and the manuscript can be accepted for publication.

Reviewer #2: (No Response)

7. PLOS authors have the option to publish the peer review history of their article (what does this mean?). If published, this will include your full peer review and any attached files.

Reviewer #1: No

Reviewer #2: No

---

## [Editor Report · Acceptance letter]

25 Jul 2023

PONE-D-22-34674R1 

Assessing Geochemical and Natural Radioactivity Impacts of Hamadat Phosphatic Mine through Radiological Indices 

Dear Dr. Zakaly:

I'm pleased to inform you that your manuscript has been deemed suitable for publication in PLOS ONE. Congratulations! Your manuscript is now with our production department. 

Kind regards, 

on behalf of

Dr. Omeid Rahmani 

Academic Editor

PLOS ONE